# Prevalence of Polyomaviruses and Herpesviruses in Moroccan Breast Cancer

**DOI:** 10.3390/pathogens12050640

**Published:** 2023-04-25

**Authors:** Amina Gihbid, Amal El Amrani, Fatima Zahra Mouh, Tarik Gheit, Mustapha Benhessou, Mariam Amrani, Sandrine McKay-Chopin, Selma Mohamed Brahim, Souha Sahraoui, Abdelouaheb Bennani, Mohammed El Mzibri, Meriem Khyatti

**Affiliations:** 1Oncovirology Laboratory, Institut Pasteur du Maroc, 1, Place Louis Pasteur, Casablanca 20360, Morocco; 2Oncogyma Research Unit, Faculty of Medicine and Pharmacy, Mohammed V University, Rabat 10100, Morocco; 3International Agency for Research on Cancer, 69007 Lyon, France; 4Department of Gynecological Obstetrics, Faculty of Medicine of Casablanca, Hospital University Center (CHU) Ibn Rochd Casablanca, Casablanca 20250, Morocco; 5Mohammed VI Center for Cancer Treatment, Ibn Rochd University Hospital, Casablanca 20250, Morocco; 6Biology and Medical Research Unit, National Centre of Energy, Nuclear Sciences and Technics, Rabat 10001, Morocco

**Keywords:** breast cancer, polyomaviruses, herpesviruses, infection, Morocco

## Abstract

Breast cancer (BC) is the most frequently diagnosed cancer and the leading cause of cancer death in women worldwide, accounting for 24.5% of total new cancer cases and 15.5% of total cancer deaths. Similarly, BC is the most common cancer among Moroccan women, comprising a noteworthy percentage of 40% of all cancers in women. Globally, 15% of cancers are attributable to infections; among them, viruses play a significant role. The present study aimed to explore the presence of a wide range of viral DNA in samples recovered from 76 Moroccan patients with BC and 12 controls using Luminex technology. The explored viruses were as follows: 10 polyomaviruses (PyVs): BKV, KIV, JCV, MCV, WUV, TSV, HPyV6, HPyV7, HPyV9, and SV40; and 5 Herpesviruses (HHVs): CMV, EBV1, EBV2, HSV1, and HSV2. Our results revealed the presence of PyVs DNA in both control (16.7%) and BC tissues (18.4%). Nonetheless, HHV DNA was detected exclusively in BC tissues (23.7%), with a predominance of Epstein–Barr virus (EBV) (21%). In conclusion, our study highlights the presence of EBV in human BC tissues, which may play an important role in its development and/or progression. Further investigations are needed to confirm the presence/co-presence of these viruses in BC.

## 1. Introduction

Worldwide, breast cancer (BC) represents a major public health problem affecting more than 2.25 million women annually and causing more than 680,000 deaths globally per year [1]. In Morocco, BC is the most common cancer and represents a real public health problem, with 11,747 new cases diagnosed and an age-standardized incidence rate of 56.4 per 100,000 women. Globally, about 630,000 deaths from BC occur annually, and close to 3700 deaths were reported in Morocco each year [2].

Numerous studies over the world have shown that BC is a multifactorial disease in which gender, age, and family history are the most important risk factors. Indeed, 99% of cases counted in the US were women and 71% were over 40 years old [3]. Moreover, 5% to 10% of BC cases are hereditary, and are due to mutations in specific genes called cancer predisposition genes, which confer varying risks of BC development: high risk (BRCA1 and BRCA2); moderate risk (TP53 and BRIP1); and finally, low risk (ATM and CHEK2) [4].

Moreover, other modifiable and non-modifiable risk factors, including greater breast tissue density, greater body mass index, physical inactivity, alcohol consumption, smoking, nulliparas, hormone replacement therapy and radiation exposure, were identified to be associated with BC development and/or progression [5,6,7,8].

Taken together, these risk factors could not explain the onset of BC for some cases and other hypotheses were explored. During the last decades, viral etiology of BC development acquired interest and many research studies have focused on identifying a potential association between viral infection and BC development and/or progression. In this field, the constant presence of viral genetic material in primary and metastatic cancer tissues (in contrast to normal tissues) and the ability of viruses to transform normal cells to cancer cells are the main arguments used to support the potential viral etiology of BC development/progression [9,10,11,12,13]. Currently, three viruses have been identified in BC samples, and their involvement in breast carcinogenesis is widely discussed: human mammary tumor virus (HMTV), homolog of the mouse mammary tumor virus (MMTV), Epstein–Barr virus (EBV) and human papillomavirus (HPV) [13,14,15,16].

Of particular interest, many studies have focused on the impact of viral co-infection in the aggressiveness of cancers. In this field, HPV was reported to potentiate EBV infection in NPC development [17] and the co-infection of EBV and HPV was reported to play an important role in the progression of cervical cancer and was found likely to be more aggressive [18]. In this regard, the studies conducted on BC could not be regarded as conclusive and more investigations are required to establish the causal association that will be used in the overall management of BC [9,19,20].

Currently, 16% of all cancers diagnosed worldwide are attributable to infectious agents and the highest burden of infection-related cancers is reported in less developed and developing countries, including countries from the North African region [21]. In Morocco, the associations between HPV and HMTV infections and BC development were already investigated. In detail, HPV DNA was found in 25.0% of BC tumors and 8.3% of controls. However, specific MMTV-like env sequences were found in 57.14% of BC tumors and 33.3% of controls. The conclusions of these studies required the need for further studies on large sampling in order to elucidate the probable causal roles of these two viruses in BC, and those of other viruses [22,23].

Thus, and as part of the international effort to inspect the viral etiology of BC, we have planned this study to investigate the presence in BC tissues from Moroccan subjects of 10 polyomaviruses (PyVs): BK-virus (BKV), Karolinska Institute virus (KIV), John Cunningham virus (JCV), Merkel Cell Polyomavirus (MCV), Washington University virus (WUV), trichodysplasia spinulosa virus (TSV), human polyomavirus 6 (HPyV6), human polyomavirus 7 (HPyV7), human polyomavirus 9 (HPyV9), and simian virus 40 (SV40); and 5 human herpesviruses (HHVs): cytomegalovirus (CMV), Epstein-Barr virus types 1 (EBV1) and 2 (EBV2), and herpes simplex virus types 1 (HSV-1) and 2 (HSV-2).

## 2. Materials and Methods

### 2.1. Sampling

In this study, a total of 76 Moroccan patients with BC were recruited from Ibn Rochd hospital (Casablanca, Morocco). These patients were randomly selected from women undergoing surgical excision for diagnosis or treatment of their breast lumps. Moreover, 12 patients with fibroadenoma were recruited as controls. To avoid any risk of potential contamination, fresh tissues were carefully collected by surgeons according to a standardized protocol using disposable instruments. Breast tissues were then stored in liquid nitrogen until further processing. Sociodemographic characteristics and clinical data were collected retrospectively from patients’ medical records. The study was approved by the ethical committee of the Pasteur Institute of Morocco (IPM/2010/31), and written informed consent was obtained from each patient.

### 2.2. DNA Extraction

Total genomic DNA was extracted using the Qiagen EZ1 Biorobot EZ1 and the EZ1 DNA tissue kit according to manufacturer’s specifications (Qiagen, Hilden, Germany). Briefly, frozen tissues were lysed in 190 µL of G2 buffer containing proteinase K (Qiagen, Hilden, Germany) at 56 °C with continuous mixing at 750 rpm on a Thermo-mixer instrument (Eppendorf, Hamburg, Germany) for at least 3 h or until the tissue was completely lysed. The extracted DNA was used immediately or stored at −20 °C until used.

### 2.3. Detection of Viral DNA

The presence of viral DNA was detected using highly sensitive type-specific PCR bead-based multiplex genotyping (TS-MPG) assays that combine multiplex PCR and bead-based Luminex technology (Luminex Corp., Austin, TX, USA), as described previously [22,24,25,26]. The multiplex PCR specific primers used for the detection of 10 polyomaviruses (BKV, KIV, JCV, MCV, WUV, TSV, HPyV6, HPyV7, HPyV9, and SV40) and the 5 herpesviruses (CMV, EBV1, EBV2, HSV1, and HSV2) are available upon request. The cut-off was calculated as previously reported by Schmitt et al. [26]. For each probe, the background values correspond to the median fluorescence intensity (MFI) values obtained in the absence of PCR products in the hybridization mixture. The cut-off was therefore computed by adding 5 MFI to 1.1 X the median background value [26,27]. In these assays, the beta-globin gene was used as a positive control and empty tubes, blindly tested, and water containing tubes were used as negative controls.

### 2.4. Statistical Analysis

Data were analyzed using the statistical software SPSS version 13.0 [28]. Statistical analyses were performed using Fisher exact chi-square test, which enables the testing of very small numbers. Differences were considered statistically significant for *p* < 0.05.

## 3. Results

### 3.1. Patient’ Characteristics

The demographic characteristics of the 76 subjects and the 12 controls showed a mean age of 38.9 and 46.9 years old, respectively. Based on clinical history and/or pathological features, of the 76 BC cases, 13 (17.1%) were diagnosed as inflammatory BC (IBC). Distribution of tumors according to the histological diagnosis showed that invasive ductal tumors prevailed and were reported in 84.2% of BC cases (64/76). Invasive ductal carcinoma combined with in situ ductal carcinoma was reported in 3.9% of cases (3/76), phyllode tumors in 3.9% of cases (3/76), ductal in situ carcinoma in 2.6% of cases (2/76), and invasive lobular carcinoma in 2.6% of cases (2/76). One case showed invasive ductal carcinoma with Paget disease (1.3%), and one case was an invasive ductal carcinoma with invasive lobular carcinoma (1.3%).

### 3.2. Prevalence of Polyomaviruses and Herpesviruses in Breast Cancer

For all samples, amplifiable DNA was confirmed by a positive PCR amplification of β-globin gene, allowing their use for viral DNA analysis.

Overall, PyVs were detected in 18.4% of BC cases (14/76) and in 16.7% of controls (2/12). In this study, among the 10 PyVs genotypes sought, only MCV, HPyV6, and HPyV7 were detected. MCV, being the most prevalent PyV detected, was identified in 14.5% of BC cases (11/76) and in 16.7% of controls (2/12) with no statistically significant difference (*p* > 0.05) (Table 1). HPyV6 and HPyV7 were not detected in controls but detected in 2.6% (2/76) and 1.3% (1/76) of cases, respectively. Statistical analysis showed no significant association between infection with these PyVs genotypes and BC development (*p* > 0.05). Of note, all PyVs-positive cases were invasive ductal carcinoma.

Of particular interest, PyVs were detected exclusively in non-IBC cases. Indeed, MCV, HpyV6, and HpyV7 DNAs were detected in 17.5% (11/63), 3.2% (2/63), and 1.6% (1/63) of non-IBC cases, respectively (Table 1). However, although PyVs were detected only in non-IBC cases, statistical analysis did not show any significant association between PyVs detection and the inflammatory status of BC (*p* > 0.05).

In this study, no HHV was detected in control specimens. However, these viruses were detected in 23.7% of BC cases (18/76). EBV was the most prevalent HHV detected in our BC cases and was detected in 21% of cases (16/76): 15.8% (12/76) for EBV-1, 5.2% (4/76) for EBV-2, and no co-infection with the two EBV genotypes was reported. One BC case was found to be positive for CMV and another one for HSV-2 (Table 2). Statistical analysis did not show any significant association between infection with HHV and EBV and BC development (*p* > 0.05).

The distribution of HHV infection according to the inflammatory status of BC is also reported in Table 2. The HSV-2 positive case was non-IBC, whereas EBV1 and CMV were detected in one IBC case each. Statistical analysis did not exhibit any significant association between infection with HHV and IBC development (*p* > 0.05).

## 4. Discussion

Worldwide, viral infections are on the rise and growing interest is given to their association with cancer development and/or progression. In this field, the etiological role of viral infection in cancer development, including BC, is gaining importance and could substantially explain the high number of sporadic BC cases [9].

Currently, great interest is given to the association between HMTV and HPV infections and BC development. Nevertheless, the low prevalence of HMTV- and HPV-BC-positive cases clearly highlight the presence of other etiologic factors of BC [23,29].

In the present study, we have investigated the presence of PyVs and HHVs in BC tissues from Moroccan patients. Results obtained showed that 18.4% of BC cases and 16.7% of controls were PyVs positives. The PyVs genotypes detected were HPyV6, HPyV7, and MCV, reported in 2, 1, and 11 cases, respectively, with no significant association with BC development. These results are in agreement with an Australian study, where MCV, HPyV6, and HPyV7 were detected in single cases (2% each) of the 54 breast tumor samples [30]. An Algerian study also detected a single MCV infection among a total of 123 BC cases [31]. However, Khan et al. showed that all BC samples examined were negative for MCV in the United Arab Emirates [32].

In the present study, HHVs were reported in 23.7% of BC cases and not in controls, EBV being the most prevalent virus found, reported in 21% of cases (18/76). These results are in agreement with widely reported data, highlighting the great association between EBV infection and BC development, thus encouraging researchers to deeply investigate the role of this virus in BC development and/or progression. In Europe, EBV DNA was found in 33.2% of BC cases with EBV infection being associated with a more aggressive phenotype [33]. In Algeria, the neighboring country, EBV was the only HHV detected in BC, with a frequency of 8.1% (10/123) [31]. Furthermore, and in a large recent meta-analysis conducted on 2402 BC cases and 1044 controls, EBV DNA was detected in 731 (30.4%) BC tissues, compared to 52 (7.5%) benign breast tissue controls (*p* = 0.001) [14]. Our data further showed that among the 16 EBV positive cancer cases identified in our study, 12 harbored EBV-1 genotype and only 4 had the EBV-2 genotype. The high frequency of EBV-1 compared to EBV2 is widely reported and discussed. An Algerian study showed that the 10 EBV positive EBV cases were all positive for EBV-1 [31].

From another point of view, the relationship between EBV and BC development may not be easy to establish due to the fact that EBV infection is nearly ubiquitous, which constrasts with the relatively rare development of BC [33]. Moreover, the difference in the prevalence of EBV in BC cases between countries may be due to the difference in the overall EBV infection in the population [14,33]. So, some researchers have focused their efforts on identifying the cellular location of EBV genetic material in BC, and they have showed that EBV DNA has a higher prevalence in cancer cells than in infiltrating lymphocytes, and that the cellular source of the PCR EBV signal came especially from the epithelial tumor cells [33,34].

Scientific evidence showed that EBV, the etiologic factor of nasopharyngeal carcinoma, is a cofactor greatly involved in the development of other malignancies, including gastric carcinomas, cervical cancer, and some lymphomas. Currently, a growing number of studies report the presence of EBV in BC tumors and consider that this virus could play an important role in breast carcinogenesis. Of particular interest, many studies have reported a synergic activity of EBV and HPV; EBV is able to potentiate the oncogenic effect of HPVs to initiate cancer pathways, and their co-presence is associated with highly aggressive tumor phenotypes in the head and neck, as well as colorectal, cervical, and breast cancers [18,35,36,37,38,39,40].

The presence of other HHVs was limited in this study to one CMV and one HSV2 positive case. In Egypt, El-shinawi et al. have reported that 60.8% of BC cases were CMV positive, suggesting that once again, the presence of these viruses in BC cases is population dependent and could be related to the overall prevalence of the viral infection [41].

In BC, the inflammatory phenotype is an uncommon and rare form, and is considered as the most aggressive form of non-metastatic BC [23]. In the current study, the inflammatory status was reported in 17.1% of cases (13/76). Of particular interest, PyVs were detected exclusively in non-IBC cases and all IBC cases were PyVs free. However, EBV1 was reported in 1 IBC case and CMV in another IBC case. To the best of our knowledge, limited studies have addressed the viral infection status in IBC cases. In Egypt, El-Shinawi et al. have shown a high prevalence of EBV in IBC cases, and also the presence of HSV 1 and 2 [41]. Interestingly, in the Egyptian population, all published results converge to the predominance of CMV in IBC cases [41,42]. In Algeria, EBV appeared more prevalent in IBC tumors (14%) than in non-IBC tumors (6%) [31].

Virus infection may contribute directly or indirectly to the specific phenotype of IBC, further investigations are essential to the underlying mechanisms of interaction and co-interaction of these oncoviruses in the carcinogenesis of this aggressive subtype of BC. Taken together, these studies highlight the presence of these viruses in BC cases, suggesting that they could play a substantial role in BC development. The differences between these studies could be due to the overall prevalence of viral infection in the whole population and the technique used for viral amplification and detection.

The present study is very informative and reports the high prevalence of EBV infection in BC cases in Morocco, supporting the possible role of EBV infection in BC development and/or promotion as an etiologic factor or co-factor in the oncogenic process that increases the risk of some BC subtypes. Furthermore, this study highlights the value in further investigating the association between EBV infection and breast carcinoma risk. However, this study presents some limitations that need to be overcome in order to achieve more consistent conclusions. These limitations consist mainly of the following: (1) the low sample size in both the BC cases and the matched controls; (2) that the assessment of viral infection according to BC subtyping to highlight the possible association between viral infection and specific BC types; (3) that the assessment of viral infection according to the tumors’ clinical–pathological features, (4) that the EBV positive BC cases were not confirmed by EBER in situ hybridization.

## 5. Conclusions

Our data revealed that EBV is present in human BC and could play an important role in its development and/or progression, indicating a clear need for further investigations to confirm the presence/co-presence of these oncoviruses in BC cases, to identify their association with specific cancer subtypes, and to elucidate the specific mechanisms involved in breast carcinogenesis.

## Figures and Tables

**Table 1 pathogens-12-00640-t001:** Prevalence of polyomaviruses DNA in breast cancer cases and controls according to the inflammatory status of cancer cases.

Virus		Controls (N = 12) n (%)	All Cases (N = 76) n (%)	OR (CI 95%)	*p* Value	IBC (N = 13) n (%)	Non-IBC (N = 63) n (%)	OR (CI 95%)	*p* Value
PyVs	Negative	10 (83.33)	62 (81.58)	1.00	-	13 (100)	49 (77.78)	1.00	-
	Positive	2 (16.67)	14 (18.42)	-	1	0 (0)	14 (22.22)	-	1
BKV	Negative	12 (100)	76 (100)	1.00	-	13 (100)	63 (100)	1.00	-
	Positive	0 (0)	0 (0)	-	1	0 (0)	0 (0)	-	1
KIV	Negative	12 (100)	76 (100)	1.00	-	13 (100)	63 (100)	1.00	-
	Positive	0 (0)	0 (0)	-	1	0 (0)	0 (0)	-	1
JCV	Negative	12 (100)	76 (100)	1.00	-	13 (100)	63 (100)	1.00	-
	Positive	0 (0)	0 (0)	-	1	0 (0)	0 (0)	-	1
WUV	Negative	12 (100)	76 (100)	1.00	-	13 (100)	63 (100)	1.00	-
	Positive	0 (0)	0 (0)	-	1	0 (0)	0 (0)	-	1
MCV	Negative	10 (83.33)	65 (85.33)	1.00	-	13 (100)	52 (82.53)	1.00	-
	Positive	2 (16.66)	11 (14.47)	0.8478 (0.1479–8.9915)	1	0 (0)	11 (17.46)	INF (0.536–INF)	0.194
SV40	Negative	12 (100)	76 (100)	1.00	-	13 (100)	63 (100)	1.00	-
	Positive	0 (0)	0 (0)	-	1	0 (0)	0 (0)	-	1
HPYV6	Negative	12 (100)	74 (97.37)	1.00	-	13 (100)	61 (96.82)	1.00	-
	Positive	0 (0)	2 (2.63)	-	1	0 (0)	2 (3.17)	INF (0.037–INF)	1
HPYV7	Negative	12 (100)	75 ()	1.00	-	13 (92.30)	62 (100)	1.00	-
	Positive	0 (0)	1 (1.31)	-	1	0 (0)	1 (7.69)	INF (0.0053–INF)	1
HPYV9	Negative	12 (100)	76 (100)	1.00	1	13 (100)	63 (100)	1.00	-
	Positive	0 (0)	0 (0)	-		0 (0)	0 (0)	-	1
TSV	Negative	12 (100)	76 (100)	1.00	1	13 (100)	63 (100)	1.00	-
	Positive	0 (0)	0 (0)	-		0 (0)	0 (0)	-	1

**Table 2 pathogens-12-00640-t002:** Prevalence of herpesvirus DNA in breast cancer cases and controls according to the inflammatory status of cancer cases.

Herpesvirus		Controls	Breast Cancer Cases						
		(N = 12) n (%)	All Cases (N = 76) n (%)	OR (CI 95%)	*p* Value	IBC (n = 13) n (%)	Non-IBC (N = 63) n (%)	OR (CI 95%)	*p* Value
Herpesviruses	Negative	12 (100)	58 (100)	1.00	-	12 (92.3)	46 (73.01)	1.00	-
	Positive	0 (0)	18 (23.68)	INF (0.763–INF)	0.115	1 (7.69)	17 (26.98)	4.37 (0.562–200.376)	0.172
EBV	Negative	12 (100)	60 (78.95)	1.00	-	12 (92.3)	48 (76.19)	1.00	-
	Positive	0 (0)	16 (21.05)	INF (0.649–INF)	0.113	1 (7.69)	15 (23.80)	2.513 (0.306–118.181)	0.678
EBV-1	Negative	12 (100)	64 (84.21)	1.00	-	12 (92.30)	52 (82.53)	1.00	-
	Positive	0 (0)	12 (15.79)	INF (0.205–INF)	0.205	1 (7.69)	11 (17.46)	2.513 (0.306–118.181)	0.678
EBV-2	Negative	12 (100)	72 (94.73)	1.00	-	13 (100)	59 (93.65)	1.00	-
	Positive	0 (0)	4 (5.26)	INF (0.099–INF)	1	0 (0)	4 (6.49)	INF (0.131–INF)	1
CMV	Negative	12 (100)	75 (98.68)	1.00	-	12 (92.30)	63 (100)	1.00	-
	Positive	0 (0)	1 (1.31)	INF (0.004–INF)	1	1 (7.69)	0 (0)	-	0.171
HSV-1	Negative	12 (100)	76 (100)	1.00	-	13 (100)	63 (100)	1.00	-
	Positive	0 (0)	0 (0)	-	1	0 (0)	0 (0)	-	1
HSV-2	Negative	12 (100)	75 (98.68)	1.00	-	13 (100)	62 (98.41)	1.00	-
	Positive	0 (0)	1 (1.31)	INF (0.004–INF)		0 (0)	1 (1.58)	-	1

## Data Availability

Not applicable.

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
