# Peer review of "Prevalence of Polyomaviruses and Herpesviruses in Moroccan Breast Cancer"

_pathogens, 2023, doi:10.3390/pathogens12050640_

Round 1

Reviewer 1 Report

Overall the manuscript is well written however there are minor suggestions that would aid the reader.

Line 40:  Please use a period and not comma: 2,25 million to 2.25 million

Lines 77-80 In Morocco, the associations between HPV and HMTV infections and BC development 78 were already investigated, and conclusions required the need of further studies on large sampling in order to elucidate the probable causal roles of these two viruses as well as other viruses in BC. A summary statistic inserted from the referenced study may aid the reader in better understanding the value of this sentence.

Line 93 In this study, a total of  [X ?] were recruited from Ibn Rochd hospital... Please insert the value.

Section 2.2. DNA extraction. As stated the DNA samples were stored at -20. It is well documented that DNA stored long-term at such temperatures may lose sensitivity with time. Please indicate the time frame (i.e. less that one month, weeks etc.). Alternatively one may indicate whether the storage buffer allows for long-term storage. If you have performed PCR analysis on fresh and stored values you can indicate whether the signal was lost and by what level. The importance of sample storage becomes important if the expected values of viral detection are low in fresh biopsy.

3.1. Patient’ characteristics.

As stated you have 76 patients in your cohort however your inflammatory BC = 13 and your invasive ductal tumors = 64 for as sum total of 77. Please clarify this discrepancy.

The authors discuss nonviral factors such as BMI, smoking etc. that could contribute to breast cancer onset or progression (lines 52-55). The author may consider adding an additional available information concerning their cohort that includes these alternative factors along with analysis viral presence and absence.  

Reviewer 2 Report

The manucript presents data regarding the presence or absence of various polyoma- and Herpes viruses, in particular the presence of the oncogenic Epstein-Barr virus (EBV) in cancerous and no-cancerous breast (carcinoma) tissue. Bonnet et al. (1999, PMID: 10451442) had published a report that "demonstrated" the presence of EBV in the tumour cells of invasive breast carcinoma by using an antibody (named 2B4) that (was known to) cross-react with an unknown cellular protein. That publication also used PCR to show that the tissues used contained EBV-DNA. That publication prompted a plethora of publications with a lot of wasted time and effort as it was ultimately shown that the tumour cells of breast carcinoam do NOT harbour the virus as is always the case for, i.e., nasopharyngeal carcinoma (NPC). See for instance, Perriogue et al. (2005, PMID: 15824148)(this reviewer is NOT associated with the Sugden group).

It  might be possible that memory B-cells harbouring EBV infiltrate breast cancer tissues, in particular those of the inflammatory type (ICB).  Given a high percentage of EBV-positive individuals in Morocco, it is somewhat surprising that none the control tisses used in this study contained detectable amounts of EBV DNA as EBV is almost always detectable in the tissues of EBV-positive individuals. The "gold-standard" to detect EBV in any tissue or cell in EBER in situ hybridization. The number of controls vis-a-vis the tumour samples appeasrs to be low.

There are two major points that need to be addressed:

1. Were the breast cancer and the control subjects EBV seropositive?

2. Were (at least some) of the EBV PCR-positive BC cases tested by EBER in situ hybridization?

Minor points:

In line 93, the number of tested individuals is missing

Line 127: reference for the SPSS method is missing

Reference 2 (line 290): "504-Morocco-Fact-Sheets.Pdf". Given that the author McKay-Chopin is affiliated with the IARC in Lyon, France, there might be a better reference for breast cancer cases/deaths in Morocco.

The references should be checked by the authos, i.e. reference 37 (line 377 "USD $329")

I feel that the authos should make a clear statement that they could not demonstrate the presence of EBV in the tumour cells of the cases analyzed as this can unequivocally be shown only by EBER in situ hybridization.

Reviewer 3 Report

1. On line 93, page 2, the number of people included in the study is missing.

2. The authors state that the primer and probe sequences are described in reference 27; however, said reference does not describe the sequences. Therefore, authors should include a table that includes the sequences of the oligos-F and oligos-R of each virus tested.

3. The description of the patients is very little descriptive. To give the manuscript greater strength, and to correlate the characteristics of viral infection with other risk factors for breast cancer, the authors should include a table with the following characteristics: housing type, education level, socio-economic level, menopausal status, number of children, and breasfeeding duration.

4. The authors did not perform a more specific correlation of EBV infection linked to tumor grade and stage. Authors should include a table that correlates EBV positivity with the following clinicopathological characteristics of patients with breast cancer: histopathological subtypes of breast cancer, lymph node involvement, estrogen receptor status, progesterone receptor status, HER2 status, molecular classification of breast cancer (luminal A, luminal B, HER2+, and triple negative.

Round 2

Reviewer 3 Report

None